# Axonal Transport as an In Vivo Biomarker for Retinal Neuropathy

**DOI:** 10.3390/cells9051298

**Published:** 2020-05-22

**Authors:** Lucia G. Le Roux, Xudong Qiu, Megan C. Jacobsen, Mark D. Pagel, Seth T. Gammon, David R. Piwnica-Worms, Dawid Schellingerhout

**Affiliations:** 1Department of Cancer Systems Imaging, The University of Texas MD Anderson Cancer Center, Houston, TX 77054, USA; xqiu2@mdanderson.org (X.Q.); mdpagel@mdanderson.org (M.D.P.); stgammon@mdanderson.org (S.T.G.); dpiwnica-worms@mdanderson.org (D.R.P.-W.); 2Department of Imaging Physics, The University of Texas MD Anderson Cancer Center, Houston, TX 77030, USA; mcjacobsen@mdanderson.org; 3Department of Neuroradiology, The University of Texas MD Anderson Cancer Center, Houston, TX 77030, USA; dawid.schellingerhout@mdanderson.org

**Keywords:** retinal neuropathy, axonal transport, neural biomarker, ophthalmoscopy, glaucoma, macular degeneration, clinical translation

## Abstract

We illuminate a possible explanatory pathophysiologic mechanism for retinal cellular neuropathy by means of a novel diagnostic method using ophthalmoscopic imaging and a molecular imaging agent targeted to fast axonal transport. The retinal neuropathies are a group of diseases with damage to retinal neural elements. Retinopathies lead to blindness but are typically diagnosed late, when substantial neuronal loss and vision loss have already occurred. We devised a fluorescent imaging agent based on the non-toxic C fragment of tetanus toxin (TTc), which is taken up and transported in neurons using the highly conserved fast axonal transport mechanism. TTc serves as an imaging biomarker for normal axonal transport and demonstrates impairment of axonal transport early in the course of an N-methyl-D-aspartic acid (NMDA)-induced excitotoxic retinopathy model in rats. Transport-related imaging findings were dramatically different between normal and retinopathic eyes prior to presumed neuronal cell death. This proof-of-concept study provides justification for future clinical translation.

## 1. Introduction

The leading causes of vision loss for individuals 40 years and older across all ethnicities are retinopathies. The National Eye Institute estimates that 11.3 million Americans will suffer from diabetic retinopathy, 4.3 million from glaucoma, and 3.7 million from age-related macular degeneration (AMD) by 2030 [1]. The estimated total annual cost of major visual disorders in the US amounts to $35.4 billion for direct medical costs and losses in productivity, added to incalculable losses of quality of life [2,3].

The retinopathies are a family of diseases that share the common hallmark feature of loss of the neural elements of the retina, for which there are currently no restorative treatments [4]. The retinal ganglion cells (RGCs) and their associated retinal axons (RAs) are well known neuronal targets of retinopathies in general, these cells being the final common pathway for information to leave the retina and travel to the brain proper through the optic nerve, which is comprised of RGC axons. Glaucoma, in particular, is known to have RGC damage as its principal feature, and the measurement of this damage has been the focus of much research. RGC damage is currently measured clinically by using visual field testing or automated perimetry, or by measuring retinal nerve fiber layer (RNFL) thickness with optical coherence tomography [5,6]. Unfortunately, these current clinical imaging modalities tend to show changes late in the course of disease when there already is substantial neuronal and vision loss [7,8].

Recognizing the deficiencies of current methods for early diagnosis and interventions, much research has been done to find better biomarkers for retinal neural health. The in vivo labeling of healthy RGCs using retrograde labeling with fluorescent dyes or tracers injected to retinorecipient areas in the brain [9,10,11] or by using anterograde labeling post intravitreal injections with fluorescently labeled tracers, like cholera toxin B (CTB) [11], have been established as research procedures but have not achieved clinical translation. Newer work tracing the molecular mechanisms causing RGC damage using in vivo TcapQ (a cell-penetrating caspase-activated apoptotic probe activated by effector caspases) can identify apoptotic cells after an intravitreal injection [12,13,14] and has potential to aid in the earlier diagnosis of glaucoma. The identification of apoptotic RGCs is currently in a phase I clinical trial (NCT02394613) using in vivo imaging techniques to detect apoptotic retinal cells after intravenous injection of Annexin-5, a marker of early apoptosis, which binds to phosphatidylserine residues in the outer leaflet of cell membrane bilayers [15].

There is an urgent need for a clinically translatable diagnostic to detect and quantify the key early molecular events in the development of retinopathy, before the onset of irreversible neuronal loss and the progression of visual loss. To address this unmet need, we sought to develop an in vivo neural imaging agent that can measure the process of axonal transport by simulating the cargo normally carried by this mechanism, like neurotrophins. In addition, there is growing evidence that disruption of axonal transport precedes presumed RGC cell death in primate retinopathy models [16,17,18] and also for glaucoma in humans [19].

Based on this evidence, we hypothesized that axonal transport is necessary for neuronal health. More specifically, damage to axonal transport results in disruption of the intricate two-way communication between RGCs and their axons that transverse through the optic nerve head to the visual centers in the brain via the optic nerve over a distance of approximately 5 cm for humans [20]. Fast anterograde axonal transport plays a crucial role in relocating proteins and mitochondria synthesized in the RGC to optic nerve synapses in the brain [21,22]. Fast retrograde transport is responsible for clearing misfolded and aggregated proteins from the optic nerve synapses and for the intracellular transport of distal neurotrophins back to the RGCs [23]. On a molecular level, axonal transport is driven by the molecular motors kinesin (anterograde) and dynein (retrograde) via specialized enzymes using ATP hydrolysis along the cytoskeletal microtubular railways [24]. Axonal transport is essential for delivering endocytosed cargo and their respective receptors packed in signaling endovesicles that undergo sophisticated sorting mechanisms involving small RabGTPase proteins. Specifically, the exchange of Rab5 for Rab7 enables fast retrograde transport of neurotrophin-containing endovesicles while Rab11 mediates the anterograde recycling of Trk neurotrophin receptors and proteins (reviewed by [25]).

TTc is an ideal molecular nerve imaging biomarker because of its known capacity to undergo neuronal cell binding and uptake into signaling endosomes sorted via small Rab GTPase proteins for undergoing fast axonal transport driven by molecular motor proteins (reviewed in [24,26]).

In this study, we demonstrate the use of a fluorescently marked molecular neural imaging probe, based on the non-toxic C-fragment of tetanus toxin (TTc). TTc has the unique attribute of uptake and fast axonal transport by neurons, undergoes rapid axonal transport, and this process is amenable to imaging in a clinically relevant quantitative and reproducible manner. We also show impairment of transport in a well-known NMDA-induced excitotoxic rat eye model of retinopathy, early in the course of disease and prior to retinal cell loss.

## 2. Materials and Methods

### 2.1. Animal Subjects

All animal experiments were approved by our Institutional Animal Care and Use Committee (00001035-RN02, 2/17/2020). Experimental procedures were performed on mature (13–17 weeks old), male Brown Norway rats, weighing approximately 250 g each, purchased from Harlan Laboratories (Indianapolis, IN, USA). A total of 30 rats were used for this study. The experimental design included each animal in a paired design, with one eye used to induce neuropathy while the other eye served as a control. The allocation for the number of animals per experiment can found in Appendix A.

Rats were anesthetized with 2% isoflurane followed by the induction of excitotoxicity in rat eyes by injecting NMDA (N-methyl-D-aspartic acid, 80 nmol/2.5 µL) into the vitreous of one eye while the contralateral control eye received a 0.01 M phosphate buffered saline (PBS) injection. This acute animal model for retinal neuropathy is well characterized and known to induce relatively uniform presumed cell death in RGCs over time [12,27], as reviewed by [28]. Forty-eight hours after the NMDA injection, the fluorescently labeled neural imaging probe was injected into the vitreous of both the retinopathic and control eyes (8 µg TTc-488 probe in 2 µL of PBS). All intravitreal eye injections were performed under a dissection microscope by penetrating the eye at the pars plana at the border of the cornea and the sclera of the eye with a 29-gauge insulin needle to a depth of no more than 2 mm. This was followed by the co-axial insertion of a 33-gauge blunt end needle attached to a 2.5-5µL Hamilton syringe, used to slowly deliver either NMDA, PBS, or the neural probe. After each injection, the needle was carefully withdrawn, and topical Neomycin and Polymyxin B sulfates and Bacitracin Zinc antibiotic ointment were applied (Bausch Lomb, Bridgewater, NJ, USA). The experimental set-up for NMDA treatment and for the TTc-488 neural probe injections can be found in Diagram S1.

### 2.2. In Vivo Imaging

In vivo image acquisition was performed using a confocal scanning laser ophthalmoscope (CSLO) (Retinal Angiograph II, Heidelberg, Germany), designed for clinical retinal imaging, with a 55° field of view lens attachment. The distribution of Alexa-labeled TTc was imaged with a baseline prior to injection followed by an imaging time point 3 h after an intravitreal injection of Alexa-488-TTc. The rats were positioned on the CSLO patient chin rest fitted with a level surface fitted with anesthesia. Prior to imaging, the pupils were dilated with 1% Tropicamide ophthalmic solution (Bausch and Lomb, Tampa, FL, USA). The corneal surface was protected during imaging with a custom made hard polymethyl methacrylate contact lens (Cantor and Nissel Limited, Brackley, UK). During CSLO imaging acquisition, the infrared reflection mode (IR, diode laser at 820 nm) was used to center the eye and focus on the retinal nerve fiber layer while the distribution of TTc-488 was captured in fluorescent angiograph mode with a blue solid laser at 488 nm with a 500-nm barrier filter. The images for quantitation were acquired as 25 aligned frames (5 images/s) at a video sensitivity of 80. ImageJ 1.52 (https://imagej.nih.gov/ij/download.html) [29] was used to average video clips to obtain a single low-noise high contrast image. Auto contrast mode was used for morphological characterization.

### 2.3. Ex Vivo Tissue Preparation

Animals were euthanized using CO_2_ asphyxiation at 3 h after the TTc-488 injection. The eyes were enucleated and fixed overnight at 4 °C in 4% paraformaldehyde to maintain the structural integrity of retinas. Retinal extractions were done in 0.1 M PBS by dividing the eyes coronally along the ora serrata and carefully brushing the retinas from the pigmented retinal epithelial layer. The curvature of the retina was managed by making four quadratic relief cuts, allowing flat mounting. Wide angle microscopy was performed on flat-mounted tissue in PBS on glass slides. The same tissue was subsequently processed for immunofluorescence histology, mounted onto permanent slides, and subjected to confocal imaging.

### 2.4. Immunofluorescence Histology

Whole mounted retinas were permeabilized in PBS with 0.1% triton on ice for 15 min followed by blocking in 1% bovine serum albumin (BSA) in PBS with 0.1% Triton for 2 h at 4 °C. RAs were detected with the primary antibody SMI32 (anti-Neurofilament H Non-Phosphorylated Mouse at 5:1000 µL, Millipore, Burlington, MA, USA, #NE-1023) in 1% BSA for 48 h at 4 °C followed by the secondary anti-mouse Alexa 594 (1:500 µL, Life Technologies, Carlsbad, CA, USA, A11005) at 4 °C for 24 h. For the specific detection of RGCs, the whole mounted retinas were blocked in 1% BSA in PBS with 0.5% Triton for 24 h at 4 °C. The primary antibody RBPMS (anti-RNA binding protein with multiple splicing, at 1:400 µL, Millipore #ABN1362) was incubated for 5 days at 4 °C followed by the secondary anti-rabbit Alexa 546 (3:600 µl; Life Technologies A11010) at 4 °C for 24 h. All immunostaining incubations were followed with 3 washes of 0.3 M PBS at room temperature. After immunostaining, retinal whole mounts were incubated overnight at 4 °C with the nuclear counterstain TO-PRO-3-Iodide (Molecular probes, Eugene, OR, USA, #T3605) at 3:1200 4 °C and permanently mounted in Prolong-antifade Gold solution (Life Technologies #P36935) prior to imaging.

### 2.5. Ex Vivo Epifluorescence Imaging

Whole retinal flat mount images with a field of view exceeding 5 mm were performed with a high-resolution epi-fluorescence zoom microscope (AxioZoom16 (AZ16), Zeiss Microscopy, Germany). The AZ16 is equipped with a PlanApoZ 0.5× objective that has a field working distance of 114 mm with a field of view ranging from 2.9 to 46 mm, with a lateral resolution (x-y direction) of 1.3–10 µm at 5.6–90× zoom magnification. Fluorescence imaging was performed with an X-cite 200DC light source with a 1.5-mm liquid light guide with suitable shift free filter sets (38 HE-GFP EX470/40 nm, EM525/50 nm and set 63 HE-mRFP EX572/25 nm, EM629/26 nm. The retinal flat mount images for both control and NMDA-treated eyes were acquired with the Zeiss AZ16 at 40× total magnification with a 1s exposure time (care was taken to equalize all imaging parameters after optimization, so as to not confound subsequent quantitation).

### 2.6. ROIs and Transect Analysis

In vivo uptake and transport of TTc-488 in the RGCs and their associated RAs were analyzed by placing a region of interest (ROI) over the entire fundus of images acquired with a CSLO (Retinal Angiograph II, Heidelberg, Germany), using ImageJ 1.52 [29] and recording the average fluorescence intensity for control and NMDA-treated eyes. 

Ex vivo retinal uptake was analyzed by placing an ROI over the wide-angle micrograph of the whole retina, using Zen 2 software. Matlab (R2016a, MathWorks, Natick, MA, USA) was used to place a similar ROI on the ex vivo epifluorescence images, and thresholding was performed to exclude regions of the retina that were damaged during dissection.

RA transection analysis was used to demonstrate the difference in the number of axonal strands observed between the control and NMDA-treated groups using Matlab (R2016a, MathWorks, Natick, MA, USA). On images of the whole retina (40× total magnification), vasculature was excluded from the confocal images using an adaptive threshold in Matlab to account for variations in the fluorescence background signal across the image. Circular transects with a 500µm radius were centered on the optic nerve head, with values along the transect recorded. A moving average fit, averaging over 25 adjacent data points, was applied to the transect data using Matlab (Natick, MA, USA), and the number of times the raw data crossed the baseline moving average was recorded as a measure of the number of axons crossing the transect line. Background fluorescence was determined by imaging a non-injected retina under similar conditions as for treated retinas (Appendix A).

### 2.7. Ex Vivo Confocal Imaging

Whole mounted retinas were optically sectioned using confocal microscopy (FV1000, Olympus Inc., Center Valley, PA, USA) to identify the RAs and RGCs within the outer synaptic (OS) layer followed by the inner synaptic (IS, including the bipolar nuclei ) and photoreceptor (PR) neural layers (Figure 1). Confocal images were acquired using both the 20×/0.85 oil UPlanSApo and 60×/1.4 oil PlanApo objectives with a FV5-LD405 laser diode (Olympus), argon ion 488 nm, HeNe G 543.5 nm, and HeNe R 633 nm lasers (Melles Griot, Albuquerque, NM, USA) as well as suitable filter sets (DM405/488/543 nm and DM488/543/633 nm) and phase imaging options.

Confocal optical scanning included the acquisition of a series of horizontal scans through the whole retinal thickness arranged as a vertical stack/z-stack by means of the FV1000 confocal analysis software (System version 4.2.1.20, Olympus Inc). Optical sections for the 20×/0.85 oil UPlanSApo were acquired at 1.63µm/slice intervals, with a typical imaging stack consisting of 80 optical sections (image size 529 × 529 µm) while the optical sections for the 60×/1.4 oil PlanApo objective were acquired with 0.47µm/slice intervals and a typical imaging stack consisting of 190 optical sections (image size 212 × 212 µm). Imaging volumes thus obtained were typically viewed as either a perpendicular cross section through the entire retina, or en face, parallel to the retinal layers.

The co-localization data comparing TTc-488-labeled RAs with the neurofilament (NF) marker, SMI32, was measured using optically sectioned confocal images (image size 529 × 529 µm), sharply focused on the retinal axons from 4 animals, with one control and one NMDA-treated eye per rat. At least 4 different confocal images per retina per treatment group (control or NMDA) were analyzed for the TTc-488 vs. SMI32 (NF marker) channels using Olympus imaging software. Pearson’s correlation coefficient values were calculated and compared with average point density plots using Matlab software.

The retinal synaptic layer fluorescence intensity ROIs for 3-D confocal imaging stacks were measured by locating individual synaptic layers within the retina using the nuclear TO-PRO-3-Iodide stain. Maximum image projections (MIPs) of a defined z-stack thickness (1.63 µm/slice) for the OS, IS, and PR layers of the retina were measured for 2 separate 3-D data sets per eye. Synaptic layer thickness for the OS, IS, and PS was determined by summing individual z-stacks per layer and multiplying by the individual slice thickness in µm. This procedure was done with results in five animals, with the average for each animal, entered as independent data points for statistical purposes.

The photo multiplier intensity and laser transmissivity were kept constant during the data acquisition for the co-localization and retinal synaptic layer fluorescence sections (Appendix A for details).

### 2.8. Statistical Analysis

Both the in vivo and ex vivo retinal analyses were compared using two-tailed paired t-tests using GraphPad Prism 7 for Windows (GraphPad Software, La Jolla, CA USA) with α ≤ 0.05 used to assign statistical significance.

## 3. Results

### 3.1. TTc is Rapidly Taken Up and Transported in Normal RGCs and RAs

The neural uptake and transport of the fluorescently labeled neural biomarker, TTc-488, was studied to understand the probe’s spatial localization to the retina after intravitreal injections in rats. In vivo imaging at 3 h after TTc injection into the vitreous demonstrates the localization of TTc-488 to the RGCs and their associated RAs. Imaging was done with a CSLO, designed for clinical retinal ophthalmic imaging (Figure 2A). The TTc-488 signal localized to the RGCs presenting as individual foci of fluorescence and also to the RAs, which were visible in a linear radial pattern. The vasculature presents as dark undulating linear structures centered on the optic nerve-head.

Ex vivo retinal flat mount preparations using widefield fluorescence microscopy confirmed the in vivo findings, (Figure 2B) clearly demonstrating RGCs as small foci in close association with RAs, which can now be seen as individual radial lines clustering around the dark center representing the optic nerve head. The vasculature in the fundus continue to be present but are less pronounced than at ophthalmoscopy.

Ex vivo confocal imaging of retinal flat mounts allowed the distinction of individual endovesicles containing TTc-488 in the RGC somas as well as in the separate RA strands as shown on an en face view (Figure 2C), while confocal optical sectioning of intact retinas illustrated TTc uptake in the dendritic inputs and the neuronal cell bodies of the RGCs, as well as some uptake in the bipolar cell axons in the retinal outer synaptic layer in a cross-section view (Figure 2D). The presence of TTc-488 in the RAs and RGCs was confirmed by ex vivo immunofluorescence (IF) microscopy. Specifically, the RNA-binding protein with multiple splicing (RBPMS), a selective cytoplasmic marker for RGCs (red) was shown to co-localize with TTc in neuronal somas. TTc-488 (green) appears as punctuate intracellular endovesicles as seen in an en face view (Figure 2C) alongside the more diffuse cytoplasmic marker of RGCs (RBPMS), which can also be observed in the cross-section view (Figure 2D). The RAs were detected with the monoclonal antibody anti-SMI32, which recognizes the non-phosphorylated neurofilament-H protein, which is a major constituent of axonal cytoskeletons. A high degree of co-localization was visually observed and quantitatively confirmed (Pearson’s coefficient = 0.7) in the RAs between the TTc-488 neural probe (green) and the neurofilament marker, anti SMI32 (red), that can be observed in the combined red/green channel of the overlay panel (Figure 2E).

### 3.2. TTc Axonal Transport in a Retinopathic Model

Axonal transport can be used as an indicator of neural health for studying retinopathic disease [30,31]. We induced retinopathy using a well-known excitotoxic N-methyl-D-aspartate (NMDA) rat eye model and used TTc-488 uptake and transport as a readout for the onset of retinopathy within 48 h of treatment. In vivo ophthalmoscopic imaging of retinopathic eyes showed a marked reduction in the radial linear pattern of RAs, with less uptake and transport of the neural biomarker. There was a sharp contrast in the appearance of normal and retinopathic eyes with CSLO imaging (Figure 1A). Ex vivo widefield fluorescence imaging confirmed these findings (Figure 1B), showing dramatically less uptake overall, and a marked reduction in the radial linear pattern and foci of uptake in individual neurons. Imaging was done with identical exposure factors and adjustments to allow for quantitative comparison. The average image fluorescence intensity for the normal and retinopathic eyes were significantly different for in vivo at 141.67 vs. 31.28 AU (*p* = 0.02) and ex vivo 6185.09 vs. 2620.68 AU (*p* = 0.03) retinal imaging data sets (Figure 1C,D), corresponding to a 4.5-fold and 2.4-fold difference for in vivo and ex vivo, respectively.

We performed retinal transect analysis to better capture the radial texture corresponding to TTc-488labeled RAs by plotting circular transects, at a constant radial distance of 500 µm around the optic nerve head (Figure 3A,B) and counted the number of crossings of the fluorescent curve through a smoothed average (moving average of order 25) plotted over a full 360 degrees. The number of crossing events was markedly reduced in neuropathic eyes vs. control eyes and were 365 ± 41.46 and 479 ± 48.41 (Mean ± SD), respectively (*p* = 0.003, Figure 3C,D).

### 3.3. Retinal Axonal Damage and Axonal Transport in a Retinopathic Model

Impaired axonal transport appears to precede morphological damage or death of the RGCs. We used the well-known axonal marker, SMI32, for staining non-phosphorylated neurofilaments (a major constituent of normal cytoskeletal axons) in RAs to distinguish retinal axonal damage and how it correlates this damage to TTc-488 axonal transport in the face of retinopathic disease. *En face* confocal imaging revealed the distinctly linear radial pattern of RAs with TTc-488 with pertinent vasculature presenting as dark thick cylinders superimposed on the axonal field (Figure 4A). Axonal TTc-488 co-localizes with SMI32 and there is a high degree of co-localization (Pearson’s coefficient 0.7 ± 0.05 (mean ± SD)) between TTc-488 (green) and the SMI32 neurofilament marker (red) in the RAs of normal control eyes. There was, however, a relative loss of RA fluorescence, and axonal sharpness in retinopathic eyes (Figure 4B) as well as a reduction in co-localization (Pearson’s coefficient 0.4 ± 0.09 (mean ± SD)) between the neurofilament marker and the neural probe (Figure 4C). These findings are due to a loss of axonal transport in retinopathic eyes, as the neurofilament cytoskeleton was still intact, showing only mild disorganization. This is further demonstrated by the average Pearson’s coefficient correlation values for control and NMDAtreated eyes being significantly different (Figure 4D).

### 3.4. Retinal Synaptic Layers and Axonal Transport in a Retinopathic Model

The retina is a complex organ and consists of well-defined neuronal and synaptic layers. Ex vivo confocal cross-section views on whole flat mount retinas allowed a quantitative assessment of TTc-488 localization to the various layers of the retina (Figure 5A). TTc-488 uptake for normal vs. retinopathic animals was significantly different in the outer synaptic (OS) layer, including the RGC layer, with a 2.4-fold difference, as well as for the inner synaptic (IS) layer with a 1.3-fold difference (Figure 5B). There was no significant difference in TTc-488 uptake for the photoreceptor (PS) layers. Furthermore, cell numbers were assessed by nuclear stained TO-PRO-3-Iodide (blue) and showed no significant differences between control and treated eyes for any layer (Figure 5C). There was a trend of an increase in the synaptic layer thickness in retinopathic eyes (Figure 5D), perhaps reflecting retinal swelling, but this did not rise to statistical significance. This result indicated that a loss of neural transport was not due to reduced cell number or presumed cell death but rather due to a functional defect in neural transport.

## 4. Discussion

Retinopathies kill neurons in the retina, but interventions to save vision require the diagnosis and monitoring of retinopathy at points in the pathogenesis preceding presumed neuronal cell death. We show proof-of-concept in an excitotoxic model of retinopathy, in that the loss of axonal transport precedes neuronal death by a clinically useful margin. This suggests that axonal transport can be used as an early marker of disease, perhaps early enough to permit interventions to save neurons and vision. This process can be monitored using clinically relevant molecular imaging techniques, and very likely could serve as a readout for the effectiveness of therapies.

Although damage to RAs and their somas are hallmark features of retinopathies in general, it is not clear if retinal axonal pathology follows or precedes RGC loss. Our own data adds to a growing consensus that axonal degeneration occurs before soma loss for neurodegenerative disease in general [32,33] and also for retinopathies like glaucoma. In glaucoma, it is thought that compartmentalized axonal self-destruction due to focal stress related to the anatomy of the cribriform plate in the optic nerve head can prevent neurotropic transport and/or blood flow and cause microglial activation [34,35]. If retinal axonal degeneration precedes RGC loss, a slowing retinal axonal transport could serve as an important early biomarker and predictor for developing retinal neurodegenerative disease. Dysfunction of anterograde and retrograde neural transport have been shown in primate (monkey, rodent) retinopathy models [16,17,18] as well as in postmortem human glaucoma [19] studies. Although RGC damage can be observed and quantified in the clinic by measuring RNFL thickness via optical coherence tomography [5,6], the RGC axonal layer cannot be differentiated from the adjacent RNFL using current imaging technologies. In our study, we show a clinically translatable in vivo imaging modality that can allow the measurement of the earliest signs of damage to the retinal ganglion axonal layer, an early marker of retinopathy, and prior to loss of RGCs or RAs.

The 50 kDa non-toxic TTc is an ideal molecular nerve imaging biomarker because of its known propensity for neuronal cell binding and uptake [36,37,38,39] and its demonstrated success in undergoing in vivo and in vitro axonal transport [40]. The carboxy terminal beta-trefoil of TTc has specific binding sites for carbohydrates, including lactose, galactose, sialic acid, and N-acetyl-galactosamine. High-affinity protein receptors have also been identified for TTc, including a 15 kDa, N-glycosylated protein [39] and the glycoprotein, Thy-1 [41]. TTc undergoes clathrin-mediated endocytosis [42] and uses the same endovesicle sorting mechanism used by neurotrophins (p75NTR, TrkB, and BDNF), which involves interaction with the small GTPase Rab7 protein [43], known to play an essential role in vesicle sorting for fast axonal transport [44]. Rab7-labeled TTc containing endovesicles undergo ATP-dependent fast axonal transport in motor neurons where a coordination between dynein–microtubule and myosin–actin microfilament interactions are essential for the optimal coordination of fast retrograde transport [40].

Comprehensive studies over the past 30 years have included the use of TTc as a fusion molecule with horse radish peroxidase (TTc-HRP) [28] and glucose oxidase (GO-TTc) [45] for illustrating transport to motor neurons (MNs) and the central nervous system (CNS). TTc has been investigated as a possible therapeutic molecule to inhibit apoptosis in the rat spinal cord MNs as a conjugate to either GFP-B-cell lymphoma extra-large (Bcl-xL-GFP-TTc) [46] or to brain-derived neurotrophic (BDNF-TTc) [47]. Recent studies focused on using fluorescently labeled TTc as a neural biomarker for in vivo imaging of axonal transport in the spinal cord to illustrate axonal transport impairment in animal models of oxaliplatin-induced neuropathy [48] and radiation-induced myelopathy [49].

In the late 1970s, TTc was explored as a histological marker for studying the neural innervation of the eye. Immunohistochemistry studies in pigeons confirmed that TTc injected to the ocular muscles (inferior and superior oblique) retrogradely labeled motor neurons in the contralateral trochlear nucleus followed by transsynaptic labeling of neurons in the ipsilateral and contralateral vestibular nuclei [50,51,52]. Intravitreal injections of TTc in rodents (rats, rabbits) resulted in the localization of the neural marker to the ventral and dorsal lateral geniculate in the brain as a result of anterograde transport (soma to nerve ends) 24 h after injection, but also to other retinorecipient areas of the brain, including the superior colliculus, pretectal nuclei, suprachiasmatic nucleus, and the medial, dorsal, and lateral terminal nuclei of the accessory optic system, 4 days after an intravitreal injection as a result of transneural (neuron to neuron) and retrograde (nerve ends to soma) transport [53]. Direct labeling of RGCs and their associated axons after an intravitreal injection of TTc has not been previously explored.

In experimental animal models, RGCs, and their associated RAs can be labeled by directly injecting neural tracers to the optic nerve or to retinorecipient areas in the brain, like the superior colliculus. Optic neural tracers include lipophilic membrane stains like dialkylcarbocyanine dye (DiI) that label nerves through passive diffusion [42]. In addition, active retrograde axonal tracers include fluoro gold (FG) [54,55] and horse radish peroxidase (HRP) [56,57] as well as for fluorescence conjugates like rhodamine-B-isothiocyanate (RITC) [58,59] and cholera toxin-B (CTB-488) [11]. The invasive nature of direct brain injections will not allow this application to progress to clinical translation. However, optical neural tracers can also be injected into the vitreous and serve as anterograde tracers for labeling RGCs and RAs in mammals. An intravitreal injection of CTB-488 resulted in the in vivo labeling of RAs within 30 min of injection and for RGCs within 24 h of injection (high magnification only) using CSLO [11]. In the current study, we show in vivo labeling of RGCs and their associated RAs within 3 h of an intravitreal TTc-488 injection in contrast with NMDA-treated eyes, which had a significantly lower (4.5 fold) uptake.

Our study is limited by relatively small numbers, but our results were statistically significant. Our results speak clearly to the primacy of axonal transport defects in the pathogenesis of the NMDA animal model we used, but this observation may not translate to all forms of retinopathy. Our work motivates further studies in other animal models of retinopathy [60]. Our study also provides motivation for clinical translation of this technology into human studies.

## 5. Conclusions

We demonstrated the use of a TTc fluorescent conjugate as a neural biomarker in the early diagnosis of retinopathy, demonstrating that abnormalities of axonal transport are a key early event in the pathogenesis of retinopathy, preceding cellular and axonal loss, and thus meeting criteria for a diagnostic that still falls within the window of potential therapeutic interventions. This methodology could potentially be adopted into clinical use.

## Figures and Tables

**Figure 1 cells-09-01298-f001:**
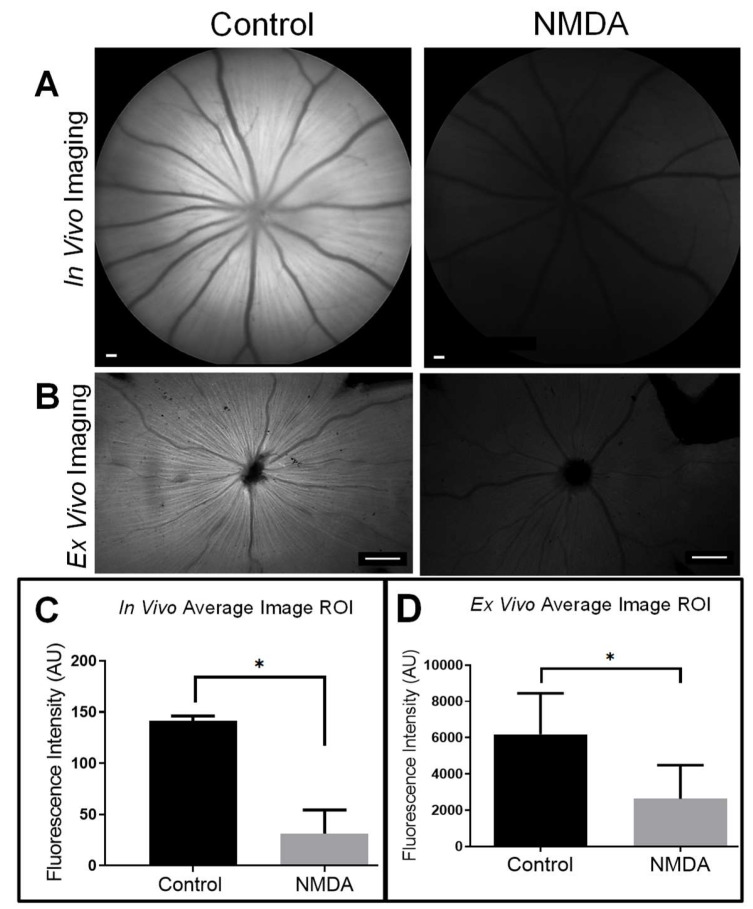
Axonal transport is impeded in a neurotoxic rat eye model. (**A**) An in vivo fluorescent image of the fundus of the eye, (**B**) and an ex vivo retinal flat mount illustrates control versus NMDA-treated eyes 3 h after an intravitreal injection of TTc-488 (8 µg in 2 µL PBS). Note the RA’s (radial linear pattern) centered on the optic nerve head in control eyes but markedly diminished in NMDA-treated eyes. The RGCs (foci pattern) are visible in the ex vivo image of control eyes only. Early retinal damage manifests as failed TTc-488 uptake and transport in retinopathic eyes. Note the close correspondence between in vivo (*n* = 3) and ex vivo (*n* = 5) quantitative findings in the (**C**) average fluorescence intensity for the in vivo control (141.67 AU ± 4.65) vs. NMDA-treated eyes (31.28 ± 23.18) (Mean ± SD) (*p* = 0.02, *t* = 7.192) and (**D**) ex vivo control (6185.09 AU ± 2262.34) vs. NMDA (2626.68 AU ± 1860.79) (Mean ± SD) (*p* = 0.03, *t* = 3.700) images compared with paired 2-tailed t-tests confirming a significant difference in TTc-488 uptake and transport between control and NMDA-treated eyes. Scale bars for in vivo 200 µm, and ex vivo imaging, 500 µm. * significant different, *p* ≤ 0.05.

**Figure 2 cells-09-01298-f002:**
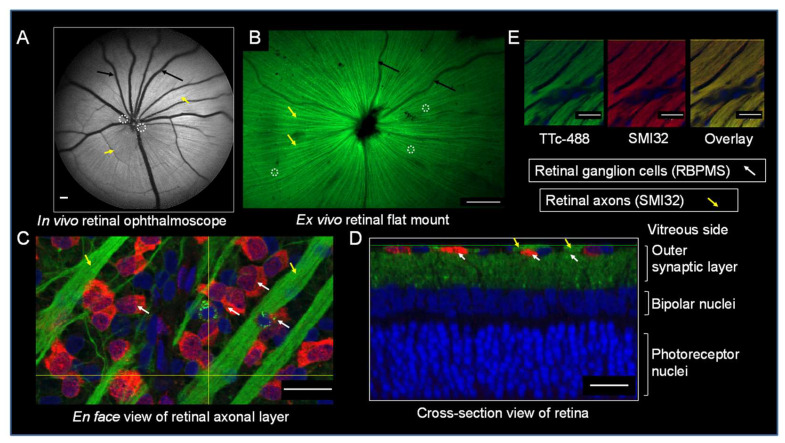
TTc-488 biomarker characterization. (**A**) An in vivo retinal ophthalmoscopic image of the eye fundus imaged with a confocal scanning laser ophthalmoscope (CSLO). Scale bar 200 µm. (**B**) An ex vivo retinal flat mount imaged with a widefield fluorescent microscope. Scale bar 500 µm. TTc-488, localizes to the RGCs presenting as small individual foci (white dashed circles) and also to the RAs in a linear radial pattern (yellow arrows) surrounding the vasculature, presenting as dark undulating lines (black arrows) centered on the optic nerve head (dark center), 3 h after an intravitreal injection. (**C**) Confocal imaging in the ganglion cell layer of retinal flat mount illustrates TTc-488 in the individual RAs arranged in bundles (green color, yellow arrows) and in RGCs (red color white arrows) as endovesicles in the perinuclear space in an *en face* view. (**D**) An optical cross-section view through the full thickness of the retina demonstrates the outer synaptic layer with extensive uptake of TTc-488 in the superficially located RGCs with their RA bundles interdigitating with bipolar dendrites, as well as the bipolar nuclei and photoreceptor nuclear layer. The localization of TTc-488 in RGCs was confirmed with anti-RBPMS-546 (red color, white arrows), a specific RGC cytoplasmic stain. **(E)** The co-localization of TTc-488 (green) in the RAs was confirmed with anti-SMI32-594 (red), reactive to a neurofilament H protein, which is a major constituent of axonal cytoskeletons. Scale bar 20 µm. TO-PRO 3 is a nuclear stain that outlines the distinct retinal nuclear layers (blue) demonstrating RGCs in the outer synaptic layer, bipolar nuclei, and photoreceptor nuclear layers. In vivo retinal ophthalmoscopic images were taken using auto contrast mode for morphological characterization of RGCs and RAs. All data in this figure were collected 3 h after a TTc-488 injection.

**Figure 3 cells-09-01298-f003:**
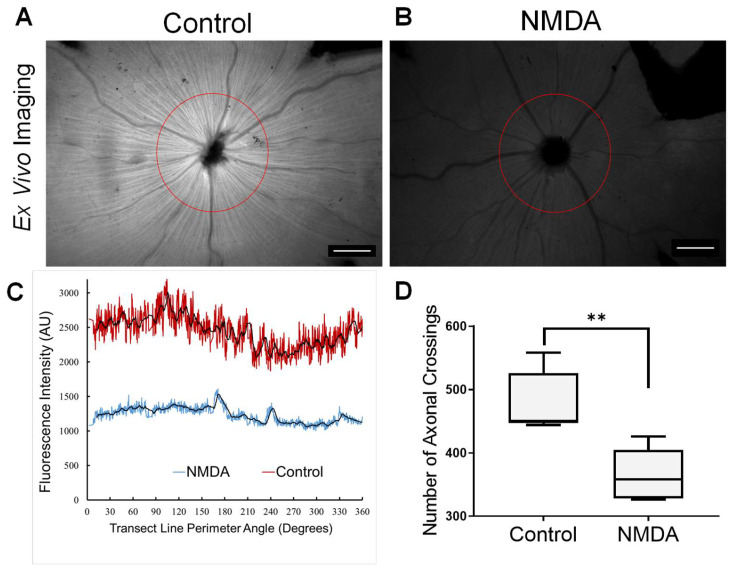
Retinal transect analysis demonstrating the impact of NMDA on axonal transport. Circular transects are drawn around the optic nerve head with a 500 µm radius (presented in circular degrees) in control (**A**) and NMDA-treated (**B**) eyes. The fluorescent intensity plotted along the transect (**C**) allows an estimate of the number of axonal strand crossings, defined as the number of times the fluorescence intensity graph crosses through a baseline moving average fit (period of 25) with vascular structures edited out. Note the greater overall magnitude of normal versus neuropathic retinas (relative average height of transects) with about 2500 AU for control and 1200 AU for NMDA-treated eyes, indicating relatively less overall uptake in NMDA eyes. The number of crossings is plotted and compared (**D**) for *n* = 5 animals and is significantly less for the NMDA-treated versus normal eyes (365 ± 41.46 versus 479 ± 48.41 crossings) (Mean ± SD) (2-tailed paired T-test, *p* = 0.003, *t* = 6.41), indicating the relative loss of the radial linear pattern representing RAs. All exposure levels kept constant for quantitation purposes. Scale bar 500 µm. ** highly significant difference, *p ≤ 0.005*.

**Figure 4 cells-09-01298-f004:**
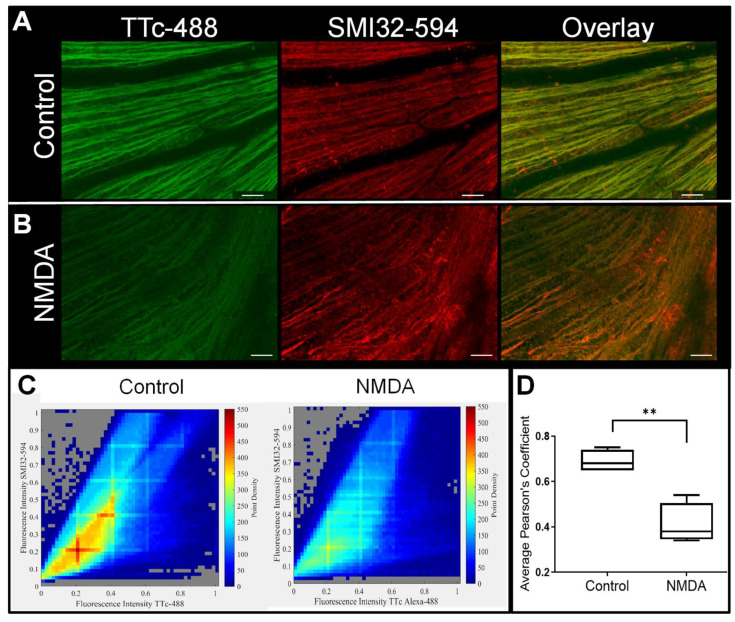
Morphological and structural changes in the retinas of NMDA-treated eyes. (**A**) The *en face* confocal view of control RAs show the highly organized radial lines of TTc-488 and SMI32-labeled neurofilaments in the RAs, with the retinal vasculature visible as dark undulations etched over this background. (**B**) The NMDA-treated eyes presented with reduced fluorescent intensity of TTc-488 and SMI32 as well as a deterioration in RA and vasculature organization. (**C**) The point density plots served as a cumulative plot for control and NMDA-treated eyes and showed that there was a high degree of co-localization for TTc-488 with SMI32 (Pearson’s coefficient 0.7 ± 0.05 (Mean ± SD)) in control eyes in comparison with the lower degree of co-localization (Pearson’s coefficient 0.4 ± 0.09 (Mean ± SD)) in NMDA-treated eyes. (**D**) There was a highly significant difference between the average Pearson’s coefficient values between control and NMDA-treated eyes (*p* = 0.0038, *t* = 1.898) for *n* = 4 animals, with 4 different imaging areas per eye for control and NMDA-treated eyes. Scale bar 50 µm. ** highly significant difference, *p ≤ 0.005.*

**Figure 5 cells-09-01298-f005:**
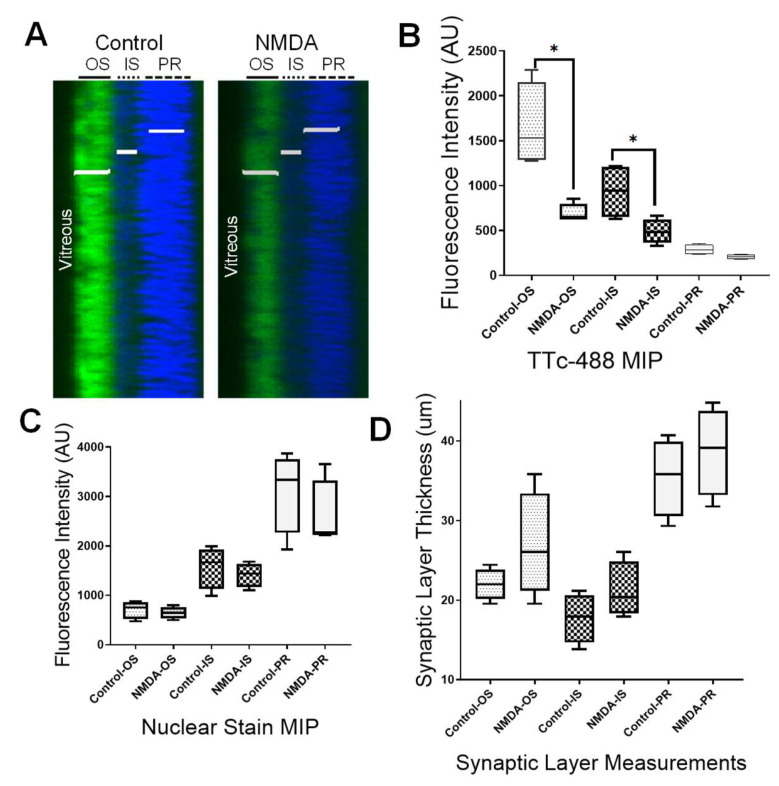
NMDA neurotoxicity affects TTc-488 uptake and transport to individual synaptic layers of the retina. (**A**) Maximum intensity projection (MIP) of a full thickness retina confocal image was obtained in a z-stack projection of 130 µm. TTc-488 fluorescence intensity was measured volumetrically for the OS layer (including the retinal ganglion cell layer), for the IS (including the bipolar nuclei) as well as the PR of the retina. (**B**) A two-tailed paired t-test between fluorescence intensity averages of control vs. NMDA-treated eyes showed a significant reduction of TTc-488 in the OS layer (1655.75AU ± 470 vs. 686.75AU ± 109, (Mean ± SD), *p* = 0.0352, t=3.660, df=3), and in the IS layer (932.38 ± 314 vs. 490.38 ± 138, (Mean ± SD), *p* = 0.0387, *t* = 3.528, df = 3), but no significant difference in the PR layer (*p* = 0.0630, *t* = 2.890, df = 3). (**C**) There was no significant difference for any of the retinal layers (OS *p* = 0.3487, *t* = 1.108; IS *p* = 0.2433, *t* = 1.448; PR *p* = 0.3786, *t* = 1.030 with df = 3 for all layers) in the nuclear-stained TO-PRO-3-Iodide (blue) fluorescence intensity, corresponding to nuclear counts, suggesting that cell number decreases were not responsible for the observed differences in TTc uptake. (**D**) The thicknesses of individual retinal synaptic layers were not significantly different between control and treated eyes at this time point in any of the synaptic layers (OS *p* = 0.2574, *t* = 1.395; IS *p* = 0.2276, *t* = 1.521; PS *p* = 0.5213, *t* = 0.7243 with df = 3 for all layers). Measurements were made on *n* = 5 animals with two imaging points per eye for all synaptic layer measurements. Scale bar 50 µm. * significant different, *p* ≤ 0.05.

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
