# Peer review of "Axonal Transport as an In Vivo Biomarker for Retinal Neuropathy"

_cells, 2020, doi:10.3390/cells9051298_

Round 1
Reviewer 1 Report
The authors describe a novel, potentially translatable test for degenerative retinopathies. The manuscript is well-written and the results are presented clearly. However, I have some major concerns regarding interpretation and extrapolation of the data.
The manuscript would benefit from a thorough proof-reading, as there are minor grammatical errors throughout. As an example:
Line 55 – there should be a comma after Annexin-5 and I’m assuming that “merger” should be “marker”.
As a general comment on the translational capacity of this test, when would this test be administered and under what conditions? In short, if this proceeds RGC death, would the subject be symptomatic in a way that would trigger this relatively invasive screening?
Line 53 – “We further hypothesized that axonal transport is a fundamentally necessary process for neuronal health”. It is well established that axonal transport is necessary for neuronal health. The second part of this statement is less of a certainty and could be conserved – “damage to axonal transport might lead to many of the findings commonly reported in retinopathy.”
Please further clarify the statement beginning on line 65 regarding ideal molecular imaging agents. Surely this is incidental to the types molecules being targeted and the imaging method used.
Please provide additional context for why an ideal marker would involve GTPAse proteins Rab5 and 7. For the uninitiated, this characteristic comes from nowhere.
References in Line 72 seem misplaced, as this statement is relating what will be demonstrated in the current study. If the fulfillment of these criteria has already been established, then the current study does not demonstrate it – the cited references do. If those references do not demonstrate it, then they are misplaced.
CSLO appears, undefined, in line 99 but is defined in line 200. In what context is CSLO currently used clinically?
As the major conclusions of this manuscript rely on the analysis of confocal images, please provide additional details regarding the confocal laser intensities and normalization schemes used in data analysis. This is briefly mentioned in the results section 3.2, but should appear in the methods and with more detail. Specifically, images can be collected with identical laser intensities, but the uptake of the dye, variation in the injection site, etc can effect the baseline fluorescence – and of course the images can be contrasted differently before/during analysis.
Figure legend 1: Is it 3 or 4 h incubation? In section 3.2, it states “within 48 h”.
Please report the fluorescence intensity values along with their standard deviations and whether these are in fact SDs or SEMS in the graphs.
The transection analysis is strange to me. How were these values of 500 um, 25 adjacent data points averaged chosen? It seems that, if you have a lower overall intensity, you will automatically detect fewer axonal crossings using this scheme and that the images should be contrasted to provide some maximum intensity value. Otherwise – since you are reporting this as an axon count - it seems that you are suggesting that the axons have degenerated and are not present in the excitotoxicity model? Is this really the case?
I am equally confused regarding the data in Figure 4. Just by observation, the images seem to overlap very well. Would the correlation coefficient not scale with the contrast of the two fluorescence lines. In other words, this a restatement of the previous point, and if you contrasted the NMDA images differently, does this effect your correlation coefficient? It should not.
The results and discussion sections state quite explicitly that your results indicate reduced axonal transport precedes cell death. While that’s, in and of itself, a controversial statement, the authors have not performed any experiments linking axonal transport to neuronal death or the margin by which it precedes cell death. In other words, you have not demonstrated that the RGC are not apoptotic at the time of your experiments. This seems to be a central claim of the paper should be amended.
You have not demonstrated or adequately discussed how this method could be translated to the clinic. For instance, are there other routes of administration that would adequately label axons and what is the persistence of the fluorescent marker?
You mention small n numbers but at one point mention that 30 animals were used for this study. My quick count does not come equal 30 animals. Please provide more detail regarding the use of animals in your study.
Author Response
Please see attachment.
The authors describe a novel, potentially translatable test for degenerative retinopathies. The manuscript is well-written and the results are presented clearly. However, I have some major concerns regarding interpretation and extrapolation of the data.
- The manuscript would benefit from a thorough proof-reading, as there are minor grammatical errors throughout. As an example:
Line 55 – there should be a comma after Annexin-5 and I’m assuming that “merger” should be “marker”.
Thanks. We corrected the minor grammatical errors.
- As a general comment on the translational capacity of this test, when would this test be administered and under what conditions?
These are early days on the road to the clinic, but the general application we foresee would be to assess patients with existing retinopathies, or at risk for developing retinopathy, undergoing novel treatments with our test as a quantitative readout to help assess such novel therapies. The precise conditions, of course, would need to be defined in future clinical trials and translational work currently under development.
- In short, if this proceeds (?preceeds?) RGC death, would the subject be symptomatic in a way that would trigger this relatively invasive screening?
All patients with RGC death eventually become symptomatic, the classic Bjerrum scotoma in glaucoma patients come to mind where a sickle- or comet-shaped blind spot usually presents in the peripheral visual field. Unfortunately, the CNS is remarkably adaptable and manages to compensate for such defects right to the edge of blindness. Patients present late, with ruined eyes, and little to be done to save their vision. We show proof-of-concept that the diagnosis of retinopathy can be established earlier, when more clinical options are still available.
The trigger for testing transport function could be legion: methanol exposure, uveitis, neurotoxic drug regimens, genetic disorders, family history, personal history, high risk conditions (such as diabetes) and so forth. The point of our work is not to establish the indications for assessing transport, but to establish proof-of-concept that it can be assessed.
- Line 53 (?63) – “We further hypothesized that axonal transport is a fundamentally necessary process for neuronal health”. It is well established that axonal transport is necessary for neuronal health.
The second part of this statement is less of a certainty and could be conserved – “damage to axonal transport might lead to many of the findings commonly reported in retinopathy.”
The underlined statement is posed as a hypothesis, and the data we present, suggests that the hypothesis is true for the animal retinopathy model we use in this study. In order to avoid confusion, the underlined statement in line 65 was restated in lines 65-81 as:
“We further hypothesize that axonal transport is a fundamentally necessary process for neuronal health, and that damage to axonal transport will result in disrupting the intricate two-way communication system between RGCs via their axons in the optic nerve and the visual centers in the brain (a distance of approximately 5cm for humans (Morgan 2004)). Fast anterograde, axonal transport, plays a crucial role in relocating proteins and mitochondria synthesized in the RGC to optic nerve synapses in the brain (Roy, Zhang et al. 2005, Millecamps and Julien 2013), while fast retrograde transport is responsible for clearing misfolded and aggregated proteins from the optic nerve synapses and for the intracellular transport of distal neurotrophins back to the RGCs (Perlson, Maday et al. 2010). On a molecular level, axonal transport is driven by the molecular motors kinesin (anterograde) and dynein (retrograde) via specialized enzymes using ATP hydrolysis along the cytoskeletal microtubular railways (Schmieg, Menendez et al. 2014). Axonal transport is essential for delivering endocytosed cargo and their respective receptors packed in signaling endovesicles that undergo sophisticated sorting mechanisms involving small RabGTPase proteins. Specifically, the exchange of Rab5 for Rab7 enables fast retrograde transport of neurotrophin containing endovesicles while Rab11 mediates the anterograde recycling of Trk neurotrophin receptors and proteins (reviewed by (Terenzio, Schiavo et al. 2017)).
- Please further clarify the statement beginning on line 65 regarding ideal molecular imaging agents. Surely this is incidental to the types molecules being targeted and the imaging method used.
We have restated lines 78-81 as:
“TTc is an ideal molecular nerve imaging biomarker because of its known ability to undergo neuronal cell binding and uptake into signaling endosomes sorted via small Rab GTPase proteins for undergoing fast axonal transport driven by molecular motor proteins (reviewed by Mendez (2014) and Surana (2018) (Schmieg, Menendez et al. 2014, Surana, Tosolini et al. 2018).”
- Please provide additional context for why an ideal marker would involve GTPAse proteins Rab5 and 7. For the uninitiated, this characteristic comes from nowhere.
Additional context provided in lines 73—78:
Axonal transport is essential for delivering endocytosed cargo and their respective receptors packed in signaling endovesicles that undergo sophisticated sorting mechanisms involving small RabGTPase proteins. Specifically, the exchange of Rab5 for Rab7 enables fast retrograde transport of neurotrophin containing endovesicles while Rab11 mediates the anterograde recycling of Trk neurotrophin receptors and proteins (reviewed by (Terenzio, Schiavo et al. 2017).
- References in Line 72 seem misplaced, as this statement is relating what will be demonstrated in the current study. If the fulfillment of these criteria has already been established, then the current study does not demonstrate it – the cited references do. If those references do not demonstrate it, then they are misplaced.
The authors are trying to point out the extensive literature on the neural uptake and the fast axonal transport mechanisms used by TTc and how it correlates with neurotrophins. The citation was moved to where the specific molecular mechanism of TTc uptake is discussed in lines 79-81.
- CSLO appears, undefined, in line 99 but is defined in line 200.
CSLO (confocal scanning laser ophthalmoscope) appeared undefined in the materials and methods section, where it was mentioned for the first time.
Thanks for pointing out this oversight. “CSLO” was defined in 2.3, line 222. The redefinition of CSLO in the results section (line 217) is not needed and was omitted.
- In what context is CSLO currently used clinically?
CSLO is used in clinical practice for retinal angiography, during which a fluorescent substance is injected intravenously, and imaged as it distributes through the retinal vascular tree (Balendra, Normando et al. 2015) However, the availability and ease of use lead to many non-angiographic applications for imaging in research labs, such as the use of CSLO for the in vivo imaging of RGC apoptosis with endogenous or exogenous markers (Higashide, Kawaguchi et al. 2006, Cordeiro, Migdal et al. 2011, Qiu, Johnson et al. 2014). Our own study uses CSLO as a convenient imaging modality in the living eye, to study a fluorescently labeled neural probe’s transport in RGC and RAs.
- As the major conclusions of this manuscript rely on the analysis of confocal images, please provide additional details regarding the confocal laser intensities and normalization schemes used in data analysis.
This is briefly mentioned in the results section 3.2, but should appear in the methods and with more detail. Specifically, images can be collected with identical laser intensities, but the uptake of the dye, variation in the injection site, etc. can effect the baseline fluorescence – and of course the images can be contrasted differently before/during analysis.
Details for confocal laser intensities were provided in Suppl. Table 2.
Supplemental Table 2: Confocal imaging laser setting.
Immunofluorescence with anti-SMI32 for co-localization studies (Fig. 4). |
|
488 nm PMT voltage |
395 V |
543 nm PMT voltage |
482 V |
633 nm PMT voltage |
389 V |
488 nm laser transmittivity |
10% |
543 nm PMT transmittivity |
50% |
Individual retinal synaptic layer thickness (Fig. 5) |
|
488 nm PMT voltage |
425 V |
543 nm PMT voltage |
482 V |
633 nm PMT voltage |
389 V |
488 nm laser transmittivity |
10% |
543 nm PMT transmittivity |
50% |
633 nm PMT transmittivity |
10% |
PMT -photomultiplier tube
The experiments for this study were performed meticulously by well-seasoned researchers. The eye injections were done by Dr. Qiu, who has more than 10 years of experience in this specific area. The immune fluorescent imaging for both the wide field fluorescent imaging and confocal imaging were performed keeping all imaging and experimental parameters, for a given experiment, exactly the same. The specific imaging settings were added to the supplemental data section (Suppl. Figs.1-3) to avoid disrupting the readability of the materials and methods section.
Please see augmentation in materials and methods Section 2.7, lines 210-212
“The photo multiplier tube (PMT) settings and laser transmissivity were kept constant during the acquisition of co-localization and retinal synaptic layer fluorescence data (see Suppl. Table 2 for details)”
Please understand that we hold ourselves to a high ethical standard. Adjusting levels differently for comparative data sets is bad form. Contrasting images differently is cheating and we do not engage in such behavior.
- Figure legend 1: Is it 3 or 4 h incubation? In section 3.2, it states “within 48 h”.
The experimental set-up was further clarified as a diagram in the supplemental data section to avoid confusion. Please refer to Suppl. Diagram 1.
The heading for Figure 1 will be changed to include the specific probe imaging time in the last sentence in line 264
Figure 1. ……………………All presented data were collected 3 hrs. after a TTc-488 injection.

Reviewer 2 Report
In this work L. G. Le Roux et al are adressing the in vivo, neural biomarker based on the axonal transport mechanism for the diagnosis and monitoring of retinopathy. They reported that the non-toxic C fragment of Tetanus toxin (TTc) is a new biomarker for the early diagnosisof retinopathy, demonstrating that abnormalities of axonal transport.
General comments:
- Please, add the retina autofluorescence signal under your conditions in the suppl. files.
- please, add the statistic data in each graph with calculations. How many animals were used for each measurments. In the methods section there is only the total number of animals.
- Could you estimate the rate of the axonal transport using your fluorescent biomarker. Please, add it in the article.
Author Response
Please see the attachment.
In this work L. G. Le Roux et al are addressing the in vivo, neural biomarker based on the axonal transport mechanism for the diagnosis and monitoring of retinopathy. They reported that the non-toxic C fragment of Tetanus toxin (TTc) is a new biomarker for the early diagnosis of retinopathy, demonstrating that abnormalities of axonal transport.
General comments:
- Please, add the retina autofluorescence signal under your conditions in the suppl. files.
Thanks. Please note changes in section 2.6, lines 180-181, as well as the addition of Suppl. Fig. 1-3, for an explanation concerning the autofluorescence signal or background signal for the in vivo and ex vivo ROI analysis as well as the ex vivo retinal transect analysis.
- please, add the statistical data in each graph with calculations.
Thanks. The (Mean ±SD) were added in all figures and updated in Fig. 2 (section 3.2, lines 288-289).
- How many animals were used for each measurement? In the methods section there is only the total number of animals.
The allocation for the number of animals per experiment can found in Suppl. Table 1. The mention of the table was also included in the material and methods, section 2.1, lines 100-101.
- Could you estimate the rate of the axonal transport using your fluorescent biomarker. Please, add it in the article.
We, unfortunately, do not currently have the data to address this question in our study. According to the literature the speed of anterograde transport varies from 0.6-5µm/sec and is highly dependent on the specific cargo (Morgan 2004) transported by protein containing signaling endosomes (Schmieg, Menendez et al. 2014, Terenzio, Schiavo et al. 2017). We have estimated the speed of retrograde transport for TTc-790 in the sciatic nerve to be approximately 12.1 mm/h (Schellingerhout, Le Roux et al. 2009) which agrees with published data on the fast-axonal transport mechanism reviewed by multiple authors (Maday, Twelvetrees et al. 2014, Surana, Tosolini et al. 2018, Surana, Villarroel-Campos et al. 2020).
References
Maday, S., A. E. Twelvetrees, A. J. Moughamian and E. L. Holzbaur (2014). "Axonal transport: cargo-specific mechanisms of motility and regulation." Neuron 84(2): 292-309.
Morgan, J. E. (2004). "Circulation and axonal transport in the optic nerve." Eye (Lond) 18(11): 1089-1095.
Schellingerhout, D., L. G. Le Roux, S. Bredow and J. G. Gelovani (2009). "Fluorescence imaging of fast retrograde axonal transport in living animals." Mol Imaging 8(6): 319-329.
Schmieg, N., G. Menendez, G. Schiavo and M. Terenzio (2014). "Signalling endosomes in axonal transport: travel updates on the molecular highway." Semin Cell Dev Biol 27: 32-43.
Surana, S., A. P. Tosolini, I. F. G. Meyer, A. D. Fellows, S. S. Novoselov and G. Schiavo (2018). "The travel diaries of tetanus and botulinum neurotoxins." Toxicon 147: 58-67.
Surana, S., D. Villarroel-Campos, O. M. Lazo, E. Moretto, A. P. Tosolini, E. R. Rhymes, S. Richter, J. N. Sleigh and G. Schiavo (2020). "The evolution of the axonal transport toolkit." Traffic 21(1): 13-33.
Terenzio, M., G. Schiavo and M. Fainzilber (2017). "Compartmentalized signaling in neurons: from cell biology to neuroscience." Neuron 96(3): 667-679.

Round 2
Reviewer 1 Report
Many thanks to the authors for carefully addressing many of the points raised during revision. However, a few key issues remain, are detailed below, and must be adequately addressed prior to publication.
12b) These data indicate that the axons are still present (we show them on confocal imaging, please see the data in Fig 5 detailing this) but were not taking up fluorescent marker, because of a failure of axonal transport.
Then the figure caption and associated text are erroneous. You are reporting a difference in the number of axonal crossings. In reality, the axons are there and crossing the circular band. However, they have not taken up the axonal transport marker. That case is made sufficiently in figure 4. The only thing being demonstrated by figure 5 is that this is an inadequate method for measuring the number of axonal crossings – since they are presumably still there and are just being undercounted. Please remove this figure and the associated text.
13) Again, we show both qualitative data (good visual overlap in Fig 4A ), and quantitative data (Pearson’s coefficients in Fig 4B). These are two ways of demonstrating the same thing… there is good overlap, our fluorescent marker is indeed in the axon, because it co-localizes with SMI32, a well-known neurofilament marker for retinal axons. Agree, scaling should not affect the correlation coefficient.
The qualitative data also suggests a high degree of visual overlap in Figure 4B, just at a diminished fluorescence intensity. I am suggesting that the correlation coefficient is highly dependent on the contrast of the image, making it (for images with a different dynamic range of intensity values) an inadequate metric for evaluating the morphological changes. You can demonstrate that this is not the case by auto-contrasting both images and re-running the correlation calculations. If the Pearson’s correlation doesn’t change, then this may be an adequate metric. If it does, it only demonstrates that this measure is highly dependent on the brightness values of the features and not the actual co-localization. Please perform this simple test and include it in the next response.
14) It follows reasonably to suppose that the defects in transport might have a causal relationship to the cell death that we know will follow
This and the associated issues can be fixed by adding the word ‘presumed’ prior to cell death throughout the manuscript. The issue is that it is stated rather conclusively throughout the manuscript that these changes prior to cell death, but cell death was not assessed. This also plays into the previous point of the hypothesis statement. If it is assured, based on previous studies, that defects in transport precede cell death, then you haven’t hypothesized it. And if it is not assured that defects precede cell death (“the cell death that we know will follow”), then you haven’t demonstrated it. The claims and the justification for the claims need to be consistent throughout the manuscript, and they are not currently.
Author Response
Please see attachment.
Many thanks to the authors for carefully addressing many of the points raised during revision. However, a few key issues remain, are detailed below, and must be adequately addressed prior to publication.
12b) These data indicate that the axons are still present (we show them on confocal imaging, please see the data in Fig 5 detailing this) but were not taking up fluorescent marker, because of a failure of axonal transport.
Then the figure caption and associated text are erroneous. You are reporting a difference in the number of axonal crossings. In reality, the axons are there and crossing the circular band. However, they have not taken up the axonal transport marker. That case is made sufficiently in figure 4. The only thing being demonstrated by figure 5 is that this is an inadequate method for measuring the number of axonal crossings – since they are presumably still there and are just being undercounted. Please remove this figure and the associated text.
Answer: Thanks for bringing up this point. I can see how the statement in 12b can be confusing. The better way to phrase this sentence would be to state:
“These data indicate that the axons are still present (we show them on confocal imaging, please see the data in Fig 5 detailing this) but these axons were not taking up and transporting the fluorescent marker as avidly as the control animals”
There is still uptake and transport of TTc-488 in the RGCs and their associated RAs for the NMDA treated eyes, but much less than in controls. We also added the retina autofluorescence signal values for the in vivo (Suppl. Fig. 1) and ex vivo ROI (Suppl. Fig. 2) and transect analysis (Suppl. Fig. 3) to emphasize that the baseline autofluorescence values are substantially lower for non-injected eyes in comparison with TTc-488 injected, control and NMDA treated eyes.
In Suppl. Fig. 3 (see page 3 of this document) we place a linear retinal transect through the middle of the optical nerve head (blue line in upper panel) to demonstrate the differences in fluorescence intensity between non-injected vs. TTc-488 injected control and NMDA-treated eyes. The fluorescence intensity of NMDA treated eyes are higher than that of non-injected retinas, which represents autofluorescence.
Please note, in the manuscript for Fig. 3 (line 291), we use retinal transect analysis to demonstrate the impact of NMDA on axonal transport and we clearly state that this method is used as an estimate of the number of axonal strand crossings (line 295) and also note that lower fluorescence intensity in NMDA treated eyes indicate relatively less overall uptake of TTc-488 (line 299).
In Fig. 4 we use the well-known neurofilament marker, SMI32, to emphasize the morphological and structural changes in the retinas of NMDA treated eyes. The purpose of this figure was to confirm that TTc-488 undergoes uptake in the RGCs but also continues to be transported to the RAs, confirmed by the SMI32 marker. Figure 4, panel B shows that TTc-488 is still present in the RAs of NMDA treated eyes but with a clearly reduced fluorescence intensity. The Pearson’s coefficient data (panels C and D) quantitively confirms what we observed in the confocal micrographs (panels A and B) for non-adjusted data.
We agree, Fig 5 is not designed to measure “axonal crossings” (that was the purpose of Fig 3). The purpose of Fig. 5 is to illustrate the differences in TTC-488 uptake to individual synaptic layers of the retina and to show that there is less TTc-488 uptake and axonal transport due to NMDA induced retinopathy, in the face of preserved layer thicknesses and cell numbers. It is very important to understand that our data cannot simply be explained by the absence of retinal neural elements. In our manuscript we clearly conclude in the text (lines 331-342) and in Fig. 5 (lines 343-359), that in:
Fig 5B TTc-488 uptake for normal vs. retinopathic animals was significantly different in the outer synaptic (OS) layer including the RGC layer, with a 2.4-fold difference, as well as for the inner synaptic (IS) layer with a 1.3-fold difference. Please note that quantification in Fig. 5 is substantially different from that of Figs. 3 (retinal transects) and 4 (Co-localization with axonal neurofilament stain, SMI32).
Fig 5C there was no significant difference in TTc-488 uptake for the photoreceptor (PS) layers. Furthermore, cell numbers were assessed by nuclear stained TO-PRO-3-Iodide (blue) and showed no significant differences between control and treated eyes for any of the retinal synaptic layers.
Fig 5D there was a mild trend of increase in synaptic layer thickness in retinopathic eyes, perhaps reflecting retinal swelling, but this did not rise to statistical significance. The key point here is that the layer thicknesses (an estimate of axonal and cellular numbers) were not significantly different, and thus cannot explain the differences in fluorescent uptake.
We conclude that loss of neural transport was not due to reduced cell number or cell death, but rather due to a functional defect in neural transport.
We respectfully disagree regarding the removal of Fig 5, because it shows new data that is especially relevant for clinical translation. Specifically, the clinical need is for interventions that alter outcomes prior to cellular death, and Fig 5 supports the point that axonal uptake and transport becomes abnormal prior to cellular loss.
The translation of basic research to the clinic requires an understanding of current imaging modalities in the clinic [1-4] and how we can enhance clinical imaging with basic research. Optical coherence tomography (OCT) is commonly used in the clinic to study retinal nerve fiber layers (RNFLs) thickness but this imaging modality cannot currently distinguish between individual synaptic layers [1, 2, 5, 6]. The use of in vivo neural probes in the clinic can potentially give us the opportunity to distinguish between different RNFLs and how this correlates with retinopathy progression.
https://business-lounge.heidelbergengineering.com/us/en/products/spectralis/spectralis/downloads/
13) Again, we show both qualitative data (good visual overlap in Fig 4A ), and quantitative data (Pearson’s coefficients in Fig 4B). These are two ways of demonstrating the same thing… there is good overlap, our fluorescent marker is indeed in the axon, because it co-localizes with SMI32, a well-known neurofilament marker for retinal axons. Agree, scaling should not affect the correlation coefficient.
The qualitative data also suggests a high degree of visual overlap in Figure 4B, just at a diminished fluorescence intensity. I am suggesting that the correlation coefficient is highly dependent on the contrast of the image, making it (for images with a different dynamic range of intensity values) an inadequate metric for evaluating the morphological changes. You can demonstrate that this is not the case by auto-contrasting both images and re-running the correlation calculations. If the Pearson’s correlation doesn’t change, then this may be an adequate metric. If it does, it only demonstrates that this measure is highly dependent on the brightness values of the features and not the actual co-localization. Please perform this simple test and include it in the next response.
Answer: This is a valid concern and a matter worthy of discussion. Fluorescence images are often manipulated prior to publication to make images visible to the human eye, but such manipulations can become distortive.
Jayme Johnson [7], for example explains how the ‘auto window’ function in most imaging display software linearly scales the full bit depth in the data to the brightest pixel on a histogram. This is a routinely applied manipulation for data display purposes. The key point to understand is that for data analysis purposes, the underlying dataset is used without change, and this is what we did. The act of windowing does not change the data, it only changes the way it is displayed. Nonetheless, we took pains to make even our windowing operations equivalent between Control and NMDA groups, to facilitate visual comparisons.
As explained by Johnson 2012, any non-linear manipulation of data should be declared, because it alters the raw data in ways that need to be controlled for. We did not do any non-linear transformations of our data. In fact, the only linear transformations of our data were windowing operations for display purposes, and these were done the same way for all groups.
Regarding Pearson’s correlation coefficients: We calculated these initially using the Olympus FV1000 software and when the volume of data made this tedious, used a Matlab script, in all cases using raw, untransformed data from the microscope. As expected, the Pearson’s r-values calculated were no different for these pieces of software. At the reviewer’s request, we now re-did this analysis, using R (ver 3.5.3) running in R-studio (ver 1.2.5001) using the tidyverse, readxl and broom packages (all by R-studio). Our first calculation was using un-adjusted raw data, our second was after a linear scaling procedure, dividing each pixel in a given dataset channel by the brightest pixel intensity in that channel (the channels being TTc488(x) and SMI32(y)). This procedure is equivalent to the ‘auto window’ procedure and represents a linear transformation of the data. Our results are given in the table below and shows that Pearson’s r (last two columns) is virtually identical for the raw and scaled data. The only minor differences are at 5 or more digits after the decimal point and likely reflect rounding differences related to the scaling procedure.
The r-values for four quadrants in a single eye were averaged, and these averages were plotted as boxplots by group to yield Fig 4B. We have reproduced new versions of Fig 4B using Raw/Unscaled and Scaled data below and show that these are substantially identical.
These data support the use of Pearson’s r as an adequate metric for our analysis in Fig 4.
Filename |
Group |
RatID |
Correlation Unscaled |
Correlation Scaled |
|
1 |
Rat 36 20x NMDA 12pm.xlsx |
NMDA |
Rat36 |
0.388043271 |
0.388043 |
2 |
Rat 36 20x NMDA 11pm.xlsx |
NMDA |
Rat36 |
0.369743021 |
0.369743 |
3 |
Rat 36 20x NMDA 2pm.xlsx |
NMDA |
Rat36 |
0.427759532 |
0.42776 |
4 |
Rat 36 20x NMDA 1pm.xlsx |
NMDA |
Rat36 |
0.407559072 |
0.407559 |
5 |
Rat 36 20x Control 9pm.xlsx |
Control |
Rat36 |
0.758517446 |
0.758517 |
6 |
Rat 36 20x Control 6pm.xlsx |
Control |
Rat36 |
0.649455715 |
0.649456 |
7 |
Rat 36 20x Control 5pm.xlsx |
Control |
Rat36 |
0.465883703 |
0.465884 |
8 |
Rat 36 20x Control 4pm.xlsx |
Control |
Rat36 |
0.73066531 |
0.730665 |
9 |
Rat 35 20x NMDA 6pm.xlsx |
NMDA |
Rat35 |
0.294612424 |
0.294612 |
10 |
Rat 35 20x NMDA 5pm.xlsx |
NMDA |
Rat35 |
0.342213848 |
0.342214 |
11 |
Rat 35 20x NMDA 3pm.xlsx |
NMDA |
Rat35 |
0.393308894 |
0.393309 |
12 |
Rat 35 20x NMDA 2pm.xlsx |
NMDA |
Rat35 |
0.325115217 |
0.325115 |
13 |
Rat 35 20x control 12pm.xlsx |
Control |
Rat35 |
0.689957817 |
0.689958 |
14 |
Rat 35 20x control 11pm.xlsx |
Control |
Rat35 |
0.773650012 |
0.77365 |
15 |
Rat 35 20x control 7pm.xlsx |
Control |
Rat35 |
0.773708451 |
0.773708 |
16 |
Rat 35 20x control 2pm.xlsx |
Control |
Rat35 |
0.606375065 |
0.606375 |
17 |
Rat 33 20x NMDA 12pm.xlsx |
NMDA |
Rat33 |
0.480264197 |
0.480264 |
18 |
Rat 33 20x NMDA 8pm.xlsx |
NMDA |
Rat33 |
0.7574134 |
0.757413 |
19 |
Rat 33 20x NMDA 3pm.xlsx |
NMDA |
Rat33 |
0.305303125 |
0.305303 |
20 |
Rat 33 20x NMDA 1pm.xlsx |
NMDA |
Rat33 |
0.624590303 |
0.62459 |
21 |
Rat 33 20x control 12pm.xlsx |
Control |
Rat33 |
0.645026024 |
0.645026 |
22 |
Rat 33 20x control 11pm.xlsx |
Control |
Rat33 |
0.840050878 |
0.840051 |
23 |
Rat 33 20x control 9pm.xlsx |
Control |
Rat33 |
0.634584492 |
0.634584 |
24 |
Rat 33 20x Control 3pm.xlsx |
Control |
Rat33 |
0.871508146 |
0.871508 |
25 |
Rat 32 20x NMDA 12pm.xlsx |
NMDA |
Rat32 |
0.054072319 |
0.054072 |
26 |
Rat 32 20x NMDA 7pm.xlsx |
NMDA |
Rat32 |
0.286355901 |
0.286356 |
27 |
Rat 32 20x NMDA 5pm.xlsx |
NMDA |
Rat32 |
0.78800981 |
0.78801 |
28 |
Rat 32 20x NMDA 2pm.xlsx |
NMDA |
Rat32 |
0.292821137 |
0.292821 |
29 |
Rat 32 20x Control 12pm.xlsx |
Control |
Rat32 |
0.739175562 |
0.739176 |
30 |
Rat 32 20x Control 11pm.xlsx |
Control |
Rat32 |
0.844257084 |
0.844257 |
31 |
Rat 32 20x Control 7pm.xlsx |
Control |
Rat32 |
0.270680324 |
0.27068 |
32 |
Rat 32 20x Control 3pm.xlsx |
Control |
Rat32 |
0.726810974 |
0.726811 |
14) It follows reasonably to suppose that the defects in transport might have a causal relationship to the cell death that we know will follow
This and the associated issues can be fixed by adding the word ‘presumed’ prior to cell death throughout the manuscript. The issue is that it is stated rather conclusively throughout the manuscript that these changes prior to cell death, but cell death was not assessed. This also plays into the previous point of the hypothesis statement. If it is assured, based on previous studies, that defects in transport precede cell death, then you haven’t hypothesized it. And if it is not assured that defects precede cell death (“the cell death that we know will follow”), then you haven’t demonstrated it. The claims and the justification for the claims need to be consistent throughout the manuscript, and they are not currently.
Answer: Upon carefully examining the most recent literature on axonal transport and neural degeneration it is evident that we currently don’t have a full grasp on whether axonal trafficking precedes neural cell death or whether neural degeneration leads to axonal transport defects. (reviewed in [8]).
I agree that it would be better to use the word “presumed” prior to referring to cell death and will change this in the manuscript where it occurred in lines 24, 62, 99, 343, 363.
- Balendra, S.I., et al., Advances in retinal ganglion cell imaging. Eye (Lond), 2015. 29(10): p. 1260-9.
- Crabb, D.P., et al., How does glaucoma look?: patient perception of visual field loss. Ophthalmology, 2013. 120(6): p. 1120-6.
- Weinreb, R.N., T. Aung, and F.A. Medeiros, The pathophysiology and treatment of glaucoma: a review. JAMA, 2014. 311(18): p. 1901-11.
- Weinreb, R.N., et al., Primary open-angle glaucoma. Nat Rev Dis Primers, 2016. 2: p. 16067.
- Harwerth, R.S. and H.A. Quigley, Visual field defects and retinal ganglion cell losses in patients with glaucoma. Arch Ophthalmol, 2006. 124(6): p. 853-9.
- Kerrigan-Baumrind, L.A., et al., Number of ganglion cells in glaucoma eyes compared with threshold visual field tests in the same persons. Invest Ophthalmol Vis Sci, 2000. 41(3): p. 741-8.
- Johnson, J., Not seeing is not believing: improving the visibility of your fluorescence images. Mol Biol Cell, 2012. 23(5): p. 754-7.
- Sleigh, J.N., et al., Axonal transport and neurological disease. Nat Rev Neurol, 2019. 15(12): p. 691-703.
